

# Subpolar Atlantic meridional heat transports from OSNAP and ocean reanalyses - a comparison

Susanna Winkelbauer[1], Isabella Winterer[1], Michael Mayer[1,2], Yao Fu[3], and Leopold Haimberger[1]

[1]Department of Meteorology and Geophysics, University of Vienna, Vienna Austria
[2]European Centre for Medium-Range Weather Forecasts, Bonn, Germany
[3]University of South Florida, College of Marine Science, Florida, USA

**Correspondence:** Susanna Winkelbauer (susanna.winkelbauer@univie.ac.at)

**Abstract.** Ocean reanalyses are potentially useful tools to study ocean heat transport (OHT) and its role in climate variability, but their ability to accurately reproduce observed transports remains uncertain, particularly in dynamically complex regions like the subpolar North Atlantic. Here, we evaluate currents, temperatures, and resulting OHT at the OSNAP (Overturning in the Subpolar North Atlantic Program) section by comparing OSNAP observations with outputs from a suite of global ocean

reanalyses. While the reanalyses broadly reproduce the spatial structure of currents and heat transport across OSNAP West and East, systematic regional biases persist, especially in the representation of key boundary currents and inflow pathways.

Temporal variability is well captured at OSNAP West, but none of the reanalyses reproduce the observed OHT variability at OSNAP East, especially a pronounced peak in 2015. This discrepancy in 2015 is traced to the glider region over the eastern Iceland Basin and Hatton Bank, where OSNAP data show a strong, localized inflow anomaly associated with the North Atlantic

Current (NAC). This signal is absent from all reanalyses as well as from independent, indirect heat transport estimates based on surface heat fluxes and heat content. Investigation of sea level anomalies and implied geostrophic currents further confirm that this mismatch is mainly driven by differences in flow structure rather than temperature anomalies alone.

Our results highlight both the value and limitations of reanalyses in capturing subpolar heat transport variability. While higher-resolution products such as GLORYS12V1 better represent circulation features, significant mismatches remain, espe-

cially in regions with sparse observational coverage. The findings underscore the need for improved observational networks and higher-resolution modeling to more accurately constrain subpolar OHT.

## 1 Introduction

The Atlantic Meridional Overturning Circulation (AMOC) is a critical component of the global climate system, redistributing heat and freshwater between the tropics and high latitudes. This circulation is characterized by the northward transport of warm

surface waters and the southward return of cooler, deeper waters, which together play a vital role in regulating regional and global climate variability. Variations in the AMOC can significantly impact Arctic sea ice (Serreze et al., 2007; Mahajan et al., 2011), sea surface temperatures (e.g., Yeager and Danabasoglu, 2014; Duchez et al., 2016) and, consequently, broader climate patterns across the Northern Hemisphere (Rahmstorf, 2024; Fox-Kemper et al., 2023; Buckley and Marshall, 2016; Jackson et al., 2015), highlighting the importance of understanding its variability and long-term changes. Anthropogenic greenhouse gas



emissions are anticipated to drive a long-term weakening of the AMOC (Collins et al., 2019; Rahmstorf, 2024), superimposed on its natural variability, which occurs across timescales ranging from subseasonal to centennial (Buckley and Marshall, 2016). Observations from the Rapid Climate Change–Meridional Overturning Circulation and Heatflux Array (RAPID-MOCHA, Rayner et al., 2011) at 26.5°N since 2004 have provided critical insights into AMOC variability on shorter timescales (Bryden et al., 2020; Srokosz et al., 2012), but this record is not yet long enough to detect long-term trends. Furthermore, AMOC

variability in the subtropical North Atlantic may differ from that in the subpolar region (Buckley and Marshall, 2016). To explore past AMOC changes, researchers have turned to indirect evidence, such as the "warming hole" or "cold blob" in the subpolar North Atlantic, a region that has cooled or resisted warming, contrary to global trends, likely due to AMOC slowing (Rahmstorf, 2024). Studies using proxies and indirect methods have suggested a possible weakening of the AMOC over the past century (Rahmstorf et al., 2015; Caesar et al., 2018; Thornalley et al., 2018), though with significant uncertainties

(Moffa-Sánchez et al., 2019). While climate models have successfully predicted global mean temperature changes, their ability to accurately reproduce past AMOC changes remains limited, with many models underestimating AMOC sensitivity and failing to simulate features like the observed cold blob (Rahmstorf, 2024; McCarthy and Caesar, 2023). Direct and sustained observations in the subpolar North Atlantic are therefore critical for capturing the structure and variability of the AMOC and refining model predictions.

The Overturning in the Subpolar North Atlantic Program (OSNAP) provides continuous observations of meridional transports of volume, heat, and freshwater from 2014 onward across the subpolar North Atlantic (Lozier et al., 2017; Li et al., 2017; Lozier et al., 2019). These observations offer valuable insights into the mechanisms and variability of the AMOC (Zou et al., 2020). The OSNAP observations also serve as a benchmark for validating ocean reanalyses (Baker et al., 2022) and climate models (Menary et al., 2020), which are critical for extending our understanding of AMOC variability beyond the observational

record.

Ocean reanalyses (ORAs) have become an essential tool for studying past ocean states, variabilities and long-term climate trends (Storto et al., 2019; von Schuckmann et al., 2020; Mayer et al., 2021b, 2022). However, their reliability depends heavily on the quality and quantity of assimilated data, with data scarcity in the deep ocean and high-latitude regions posing significant challenges. Despite these limitations, ORAs have been shown to realistically capture observed trends and variability in northern

high-latitude ocean heat content (OHC) (Mayer et al., 2021b). However, their performance deteriorates in data-sparse areas like the deep ocean, where observational constraints are limited (Palmer et al., 2017). Although oceanic heat transports (OHT) play a critical role in the climate system and reanalyses would provide a vital tool for studying their variability prior to the availability of direct observations, their validation has received comparatively less attention than the validation of state quantities such as OHC. As reanalyses generally do not assimilate direct observations of ocean currents, their transport estimates depend largely

on model dynamics and parameterizations rather than observational constraints. An additional difficulty is the methodological complexity of OHT estimation (e.g., depth vs. density space, reference temperature choice), which complicates validation.

Overall, past studies have demonstrated that while ocean reanalyses generally capture the mean and variability of key features such as integrated transports reasonably well, notable biases persist, particularly in the representation of deep water masses, overflow waters, and the spatial structure of currents in narrower sections (Mayer et al., 2023; Fritz et al., 2023). Jackson et al.



(2016, 2019) have shown that reanalyses can capture key aspects of AMOC variability at the RAPID array in the subtropical North Atlantic. At subpolar latitudes, Baker et al. (2022) found that reanalyses broadly reproduce the magnitude and variability of the MOC at OSNAP, supporting their use in studying the overturning circulation. However, their analysis did not assess heat transport variability or its spatial distribution, leaving important gaps in our understanding of how reanalyses represent heat transport at OSNAP.

The present study addresses this critical gap by comparing observational and reanalysis-derived OHT estimates across the western and eastern branch of the OSNAP section. These are accompanied by detailed analyses of cross-sections of currents and temperatures to find reasons between the arising differences such that we can reliably assess how well reanalyses replicate observed variability and show the value of integrating observational and model-based approaches to advance climate research.

## 2 Data and Methods

OSNAP is a sustained, international ocean observing initiative established in 2014 to provide comprehensive measurements of the Atlantic Meridional Overturning Circulation (AMOC) in the subpolar North Atlantic (Lozier et al., 2017). While the program's primary objective is to quantify AMOC variability, it also provides integrated meridional heat and freshwater fluxes. The OSNAP array, which consists of moorings, gliders and floats, includes two legs: OSNAP West, spanning from Labrador Shelf to West Greenland, and OSNAP East, extending from East Greenland to Scotland. Figure 1 illustrates the OSNAP array,

including its network of moorings and glider deployments (for further details, see Lozier et al., 2017).

We compare those observational estimates of oceanic transports to monthly output from a set of global ocean reanalyses, including ORAS5 (Ocean ReAnalysis System 5, Zuo et al., 2019), CGLORSv7 (Centro Euro-Mediterraneo sui Cambiamenti Climatici Global Ocean Reanalysis System, Storto and Masina, 2016), GLORYS2V4 (Global Ocean Reanalysis and Simulation, Garric and L.Parent, 2016), GloRanV14 (an improved version of GloSea5, also known as the FOAM, herafter called

FOAMv2; MacLachlan et al. 2015), and GLORYS12V1 (Lellouche et al., 2018). The first four products can be considered as an updated version of the Copernicus Marine Service (CMEMS) Global Reanalysis Ensemble Product (GREP; Desportes et al. 2017), while GLORYS12V1 extends the analysis with its higher resolution. The reanalyses are all based on the NEMO ocean model, with the GREP products being configured at a 0.25° horizontal resolution and 75 vertical levels, while GLORYS12V1 has a finer horizontal resolution of 0.083° and 50 vertical levels. The necessary atmospheric forcing for all ORAs was taken

from the ERA-Interim (Dee et al., 2011) atmospheric reanalysis, after 2019 from ERA5 (Hersbach et al., 2020). While this forcing is similar for all the ORAs, they differ in their data assimilation methods, physical parameterizations, and initial conditions, which contribute to important differences in their transport estimates. Notably, ORAS5 assimilates sea level anomalies (SLA) only between 50°S and 50°N, whereas the other four reanalyses assimilate SLA in ice-free seas globally. While all the considered ORAs assimilate in-situ profiles of temperature and salinity, they do not assimilate observations of currents, i.e.

they can be considered independent of OSNAP in that regard.

To assess geostrophic circulation anomalies at the surface, we use satellite-derived sea level anomalies (SLA) from the CMEMS multi-mission product (SLA). This dataset provides global, gridded SLA fields referenced to the 1993–2012 mean,





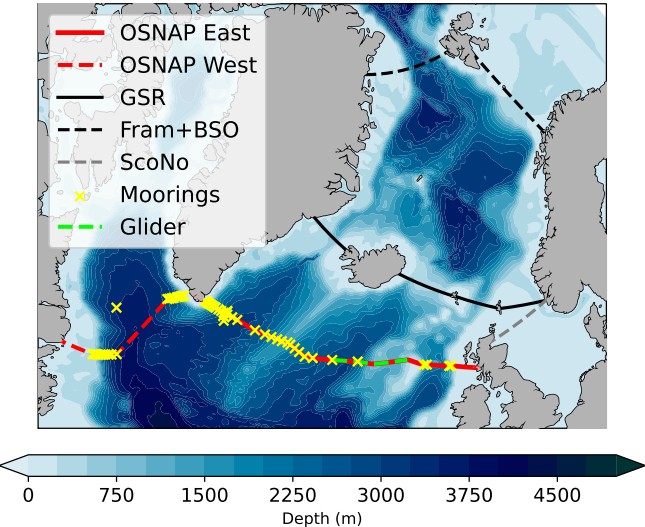

**Figure 1.** Map of the OSNAP region and the general ocean bathymetry, schematically depicting OSNAP East, OSNAP West, mooring and glider locations used for the OSNAP obseravtions, as well as the Greenland-Scotland Ridge, Fram Strait and the Barents Sea Opening (BSO).

with a 0.25° horizontal resolution. It is based on delayed-time altimeter observations from multiple satellite missions and includes standard corrections for tides, atmospheric pressure, and instrumental effects. The resulting fields allow for the esti-
mation of surface geostrophic velocities through sea level gradients.

To reduce short-term variability and highlight interannual changes, all time series are smoothed with a 12-month running mean, unless otherwise indicated.

## 2.1   Cross-sections and transports

Velocity and temperature sections reveal oceanic flow and temperature distributions across specific straits or regions, facilitating
the understanding of the structure and variability of OHT.

Fu et al. (2023) provide gridded sections of velocities, temperatures, and salinities as well as integrated transports of heat, freshwater and volume across the OSNAP line using in situ measurements from moored instruments deployed at multiple depths across the subpolar North Atlantic. These moored observations are supplemented by gliders and Argo floats, which fill spatial gaps between moorings and provide additional data in regions without moored observations. The positions of moorings
and glider sections are shown in Fig. 1. Detailed descriptions of the instruments, as well as calculation and interpolation techniques used, can be found in Li et al. (2017) and Lozier et al. (2017).

We aim to compare the observed velocity and temperature sections, as well as integrated transports across the OSNAP section, to those calculated from ocean reanalyses. Extracting accurate velocity sections and calculating transports from reanalyses is challenging as ocean models often use complex curvilinear or unstructured grids, necessitating the accurate handling of
varying grid geometries. To address this, we use StraitFlux (Winkelbauer et al., 2024), a Python tool specifically designed to





calculate cross-sections and transports in a manner consistent with the discretization schemes of the analyzed models, and to preserve conservation properties of the model's native grids. More details on the calculation methods can be found in Winkelbauer et al. (2024).

To assess cross-sectional biases and RMSE values between OSNAP and reanalyses, the reanalyses sections are interpolated bilinearly onto the OSNAP grid.

In addition to cross-sections, we analyze integrated oceanic transports through the OSNAP section. In general, oceanic transports of volume (OVT), heat (OHT), and other tracers through a given cross-section are key metrics for understanding ocean circulation and energy transfer. These transports are defined mathematically as integrals over the cross-sectional area, incorporating the velocity field and additional scalar properties such as temperature or tracer concentration. In this study, we focus on the transport of heat, which can be expressed as:

$$OHT = c_p \rho \int\limits_{x_l}^{x_r} \int\limits_{z_1(x)}^{z_2(x)} (\theta(x,z) - \theta_{ref}) \boldsymbol{v}(\boldsymbol{x},\boldsymbol{z}) \cdot \boldsymbol{n} \, dz \, dx \tag{1}$$

with the velocity vector $\boldsymbol{v}$ and the unit normal vector to the cross-section $\boldsymbol{n}$. The cross-sectional area of the strait is defined by its width $x$ and depth $z$. The density of seawater is represented by $\rho$ and the specific heat capacity of seawater by $c_p$. For simplicity, transport calculations based on reanalyses are performed assuming constant $\rho$ ($1026 kg/m^3$) and $c_p$ ($3996 J / kgK$), while for OSNAP observations transports $\rho$ and $c_p$ are calculated for each grid cell as a function of temperature, salinity, and pressure (Li et al., 2017). However, the impact of these differences should be minimal, as variations in $\rho$ and $c_p$ tend to compensate each other, resulting in negligible changes to the total transports (Fasullo and Trenberth, 2008). Additionally, for heat transports the potential temperature $\theta$ and a reference temperature $\theta_{ref}$ are needed. Transports have to be calculated relative to a reference temperature, as unambiguous heat transports require closed volume transports, which is not the case for individual straits and only approximately true for total oceanic transport (Schauer and Beszczynska-Möller, 2009). To simplify the analysis and align with previous studies the reference temperature $\theta_{ref}$ is commonly set to 0°C. To avoid interpolation and preserve conservation properties net integrated transports from reanalyses are calculated by employing StraitFlux's line integration method.

All transport calculations in this study are based on monthly mean output from the ocean reanalyses. To evaluate the potential influence of temporal resolution, we additionally tested calculations based on daily velocity and temperature fields for GLORYS12V1. The resulting differences in integrated heat transports were found to be less than 0.5% when averaged over the OSNAP period, indicating that monthly output provides a sufficiently accurate representation for the purposes of this study (see Fig.A2).

### 2.2 Indirect estimation of heat transports

To complement the ORA-based OHT estimates, a largely independent heat transport estimate was additionally obtained following the oceanic heat budget approach outlined in Mayer et al. (2023).



$$OHT_{OSNAP_E} = OHT_{GSR/FS+BSO} - \left[ F_S - \rho_0 c_p \frac{\partial}{\partial t} \int_0^Z (\theta_o - \theta_{ref}) dz - L_f \rho_i \frac{\partial d_i}{\partial t} - R_{Adj} \right]_{OSNAP_E}^{GSR/FS+BSO} \tag{2}$$

The vertical net surface energy flux, counted positive if downward, is denoted as $F_S$. We define gridpoint ocean temperatures as $\theta_o$, the reference temperature as $\theta_{ref}$, the sea ice density $\rho_i$ (assumed constant at 928 kg m$^{-3}$) and gridpoint average sea ice
thickness $d_i$. This method infers OHT at OSNAP East by integrating $F_S$, the heat content changes (OHCT; second term inside the brackets of Eq. 2) and the melt ice tendencies (MET; third term inside the brackets) over the area between the OSNAP-East section and a nearby section where observations are available.

$F_S$ is estimated indirectly from atmospheric budgets, so these are much better constrained by independent observations than parameterized surface fluxes, which typically are more uncertain and depend on the sea state (Mayer et al., 2023; Trenberth
et al., 2019). Therefore, divergences and tendencies from atmospheric reanalyses ERA5 (mass-consistent energy budgets, Mayer et al., 2021a), MERRA2 (Gelaro et al., 2017) and JRA55 (Kobayashi et al., 2015) are combined with top-of-atmosphere (TOA) fluxes from CERES-EBAF TOA version 4.2 (Scott et al., 2022; NASA/LARC/SD/ASDC, 2025).

OHC is taken from the GREP reanalyses mentioned above, the Institute of Atmospheric Physics version 4 (IAPv4 Cheng et al., 2024) and Random Forest Regression Ocean Maps (RFROM Lyman and Johnson, 2023). The melt ice tendencies MET
is calculated from GIOMAS (Zhang and Rothrock, 2003) and ORAS5, which both provide ice thickness fields at 1 degree and 1/4 degree resolution, respectively.

Mass-consistent heat transport estimates are used at two different choke-points: the Greenland-Scotland Ridge (GSR), and the combination of Fram Strait (FS) and the Barents Sea Opening (BSO). While the GSR transports are available continuously from 1993 to 2021 (Tsubouchi et al., 2021), the Fram Strait and BSO estimates are limited to the ArcGate campaign period
from October 2004 to April 2010 (Tsubouchi et al., 2024). To enable comparisons over the extended OSNAP period, we construct a synthetic ensemble of Fram + BSO transports by extrapolating the limited ArcGate observations. The extrapolation uses a first-order autoregressive [AR(1)] model that preserves the observed seasonal cycle and reproduces realistic variability based on the lag-1 autocorrelation and the standard deviation of monthly anomalies. We generate 1000 synthetic realizations to sample the range of plausible interannual variability. Using the GSR as a boundary introduces larger uncertainties due to the
relatively high magnitude and variability of the heat transport across the ridge. In contrast, integrating from Fram Strait and the BSO benefits from lower uncertainties in the upstream OHT, but requires a larger integration domain (potentially leading to larger accumulated errors) and more careful treatment of sea ice processes and storage. By considering both configurations, we leverage the strengths of each approach and account for complementary sources of uncertainty in estimating the subpolar heat transport.

An approach like this would require a closed volume. However, the region bounded by the choke-points and OSNAP East includes an open passage between Scotland and Norway (ScoNo in Fig. 1). Since exchanges through this passage are negligibly small (ORAS5 gives a mean transport of $3 \pm 8$ TW, more than two orders of magnitude smaller than transports across OSNAP East), they are excluded from the calculation. Additionally, to address global inconsistencies between air–sea heat fluxes and



ocean heat content tendencies (OHCT), we apply a spatially uniform adjustment $R_{Adj}$ to the net heat fluxes. Specifically, we
subtract the monthly difference between the global full-depth OHCT and surface fluxes at every grid point as implemented e.g.
in Mayer et al. (2023) and Trenberth et al. (2019).

To account for uncertainty in each component of the budget, we combine three estimates for surface fluxes, six estimates for
OHC, and two estimates for sea ice in all possible permutations, resulting in 36 estimates for both choke-points. The data used
is summarized in Table 1.

| variable | datasets |
|---|---|
| **Fs inferred** | ERA5, JRA55, MERRA2 |
| **OHCT** | IAPv4, RFROM, ORAS5, CGLORS, GLORYS2V4, FOAMv2 |
| **MET** | GIOMAS, ORAS5 |

**Table 1.** Datasets used for indirect heat transport estimates.

A similar approach has recently been applied and evaluated by Meyssignac et al. (2024), who used surface fluxes and heat
storage terms between OSNAP and RAPID to estimate heat transports at the RAPID section. Their results show good agreement
with direct in situ estimates in terms of the mean and reasonable agreement in terms of temporal variability, supporting the use
of energy budget methods to bridge observational gaps between ocean sections. Our analysis, which additionally incorporates
independent ocean reanalyses and explicit uncertainty estimates for all budget components, might provide a useful complement
and help to build confidence in this approach.

## 3  Results

### 3.1  Temperature and Velocity Cross-Sections

The temperature distribution along the OSNAP section is shown in Figure 2a, while the corresponding temperature biases
for the reanalyses relative to the OSNAP observations are presented in Figures 2b–f. The OSNAP temperature cross-section
reveals a distinct thermal structure across the North Atlantic, with warmer waters concentrated in the Rockall Trough and the
upper layers of the Iceland Basin, where the North Atlantic Current transports heat from lower latitudes northward. Below
the warm Atlantic Water layer, temperatures gradually decrease and transition to deeper, colder water masses. In contrast, the
Irminger Basin and the Labrador Basin are characterized by significantly colder waters. These basins are influenced by waters
of subpolar and Arctic origin, with the coldest and densest overflow waters formed in the Nordic Seas. While the reanalysis
models broadly reproduce the large-scale thermal structure observed by OSNAP, they exhibit systematic and regionally varying
temperature biases. Temperatures are predominantly underestimated in the basin interiors, with cold biases reaching up to -
1.0°C, particularly in ORAS5. In contrast, distinct warm biases appear at the Labrador Shelf as well as the West and East
Greenland Shelfs, the highly dynamical Reykjanes Ridge, intermediate waters in Icealnd basin, west of Hatton and Rockall
Bank, Rockall Trough and deep waters below 2000m along the continental slopes of the Irminger, Labrador and to a lesser





extent also Iceland Basins. While we treat OSNAP observations as a reference, comparisons with the EN4 objective analysis
dataset (Good et al., 2013) reveal some differences (see Fig. A1). EN4 generally shows a cold bias relative to OSNAP, with
particularly pronounced discrepancies near the Reykjanes Ridge and along the Labrador Shelf. These differences might reflect
EN4's coarser resolution and the sparser observational coverage in the subpolar North Atlantic, but at the same time, while
OSNAP offers more detailed and spatially resolved data, it may also be subject to its own observational and methodological

uncertainties (Li et al., 2017).

Figure 3 illustrates the temporal RMSE values relative to OSNAP observations. All reanalyses exhibit similar spatial patterns
of error, suggesting a shared underlying structure in their deviations from observations. The strongest RMSE values are found
in surface waters, particularly over the continental shelves and along the Reykjanes Ridge, as well as in intermediate waters
within the Iceland Basin, where important inflow branches of the North Atlantic Current (NAC) are located. However, no direct

mooring observations are available in this region, and ARGO (Jayne et al., 2017) floats provide the primary observational
coverage. Additionally, RMSE values are elevated below 2000 m in the Labrador Basin, where a distinct "break" in the error
pattern appears. This could be attributed to a lack of observationally constrained variability at these depths in the ORAs as
ARGO data are only available down to 2000 m.

Figure 4a depicts the time-mean currents across the OSNAP-East and OSNAP-West sections, while Figures 4b–f illustrate

the mean biases of the reanalyses relative to observations. Although the reanalyses generally capture the overall flow structure
and main currents, discrepancies exist in both the strength and exact positioning of these currents.

Observations indicate that the Labrador Current (LC) originates close to the shore along the Labrador Shelf and extends
along the continental slope down to the seafloor. In contrast, reanalysis data suggest a slightly more offshore and shallower
LC position. Additionally, the reanalyses show a weak northward recirculation east of the LC, a feature absent in OSNAP

observations, likely due to a lack of mooring coverage in that region. Similarly, all 0.25° reanalyses underestimate the strength
of the West Greenland Current (WGC) and displace it slightly towards the shore, while the East Greenland Current (EGC)
appears too broadly extended offshore, leading to an underestimation of southward current velocity along the continental slope
and an overestimation further into the basin. These biases may contribute to the positive temperature anomalies observed along
the continental slopes of the Irminger and Labrador basins. GLORYS12V1 shows very strong but very narrow velocity peaks, it

overestimates the WGC and EGC at their peaks along the continental slopes but underestimates their strength farther offshore.
Despite this sharp structure, the spatial mean absolute error remains comparable to that of the other reanalyses (see Fig. 4). The
East Reykjanes Ridge Current is systematically underestimated in all reanalyses, while the North Atlantic Current just west
of Hatton Bank is overestimated across all ORAs except GLORYS2V4. Moreover, the reanalyses exhibit pronounced inflows
and outflows within the Iceland Basin and, to a lesser extent, the Rockall Trough, regions where no mooring observations are

available, making it difficult to assess the accuracy of these features.

Figure 5 presents the velocity temporal RMSE values relative to OSNAP observations. All reanalyses display similar spatial
RMSE patterns, with GLORYS12V1 exhibiting greater variability due to its higher horizontal resolution. The largest variances
are associated with major currents, particularly the North Atlantic Current in the Iceland Basin, where mooring observations





**Figure 2.** a) Cross sections of 2014/08 to 2020/06 average temperatures along the OSNAP section; b)-f) Biases of temperatures compared to OSNAP for the respective reanalyses. Mean Absolute Error (MAE) is indicated in the top-right corner of each bias panel.

are absent. In the basin interiors, the RMSE fields exhibit a streaky appearance, which likely reflects current misplacements

but may also partly result from the sparse spatial coverage of the OSNAP moorings in the interior.

Despite broadly reproducing the thermal and dynamic structure along the OSNAP section, the reanalyses exhibit systematic biases with respect to OSNAP in both temperature and velocity fields. The cold biases in basin interiors, warm biases along continental slopes, and discrepancies in current positioning suggest limitations in how these models represent observed oceanic features given the relatively limited observational constraints. Notably, ocean currents are not directly assimilated in these

reanalyses. While the general patterns of error are consistent across reanalyses, the absence of mooring observations in key regions, such as the Iceland Basin and Rockall Trough, adds further uncertainty.



**Figure 3.** Cross-sections of temporal temperature RMSE (annual cycle removed) relative to OSNAP East for each reanalysis product over the period 08/2014 to 06/2020. Mean RMSE values are shown in the top-right corner of each panel.




**Figure 4.** a) Cross-section of 2014/08 to 2020/06 average velocities along the OSNAP section; b)-f) Bias of velocity compared to OSNAP for the respective reanalyses. Major currents are indicated: Labrador Current (LC), West Greenland Current (WGC), East Greenland Current (EGC), Irminger Current (IC), East Reykjanes Ridge Current (ERRC) various branches of the North Atlantic Current (NAC; labeled in grey where no moorings are present). Mean Absolute Error (MAE) is indicated in the top-right corner of each bias panel.





**Figure 5.** Cross-sections of temporal velocity RMSE (annual cycle removed) relative to OSNAP East for each reanalysis product over the period 08/2014 to 06/2020. Mean RMSE values are shown in the top-right corner of each panel.



## 3.2 Heat transports

Table 2 shows average heat transport values over the 08/2014 to 06/2020 period. OSNAP observations show a net heat transport
of $417 \pm 27$ TW for OSNAP East. While reanalyses generally yield lower values, all except ORAS5 fall within the observa-
tional uncertainty range. For OSNAP West, observations indicate a heat transport of $85 \pm 6$ TW. Among the reanalyses, only
the higher-resolution GLORYS12V1 (92 TW) falls within the observational uncertainty range, whereas ORAS5 significantly
overestimates it, and the remaining three reanalyses substantially underestimate the observed values. Figure 6 presents the
net integrated heat transports across OSNAP West (a) and East (b), comparing OSNAP-derived heat transports with direct
transport estimates derived from the reanalyses using StraitFlux (Winkelbauer et al., 2024). In addition to individual ORA
outputs, the GREP mean (mean of the four 0.25 ° reanalyses) is shown, along with its associated uncertainty ($\pm 1$ standard
deviation, blue shading). At OSNAP West, heat transport variability is generally lower compared to OSNAP East, though there
is a considerable spread among ocean reanalyses. The reanalyses generally capture heat transport variability well, showing
high correlations with observations (see Table 3), except for ORAS5 (r=0.12, p=0.31). GLORYS12V1, with higher resolution,
closely follows observations (r=0.79, p<0.01), while the GREP mean, though biased low, represents variability equally well
(r=0.79, p=0.01). In contrast, at OSNAP East, reanalyses generally exhibit weaker MHT variability than the observation, in par-
ticular they miss the pronounced 2015 maximum, the subsequent multi-year decline to the 2019 minimum, and the 2019–2020
increase. Consistent with this muted variability, correlations with observed MHT are low for the 0.25° reanalyses, whereas
the higher-resolution GLORYS12V1 exhibits a comparatively stronger correlation (r=0.37, p<0.01), suggesting that increased
spatial resolution may enhance the representation of heat transport variability.

The lower panel (6c) presents indirect estimates of heat transport using the GSR choke-point (blue lines) and the Fram
+ BSO choke-point (green lines), as described in Section 2.3. Thin lines show all possible combinations using the datasets
listed in Table 1. Uncertainties are estimated by combining the spread (measured as standard deviation) across the datasets
involved, the observational uncertainties associated with the GSR, and the spread across the 1000 Fram+BSO realizations.
Corresponding mean values are given in Table 2. They are biased low relative to observations, with larger uncertainties for the
Fram+BSO estimate. These uncertainties are dominated by the wider integration area, which leads to larger uncertainties in Fs
and OHCT (std = 51 TW), whereas the contribution from the synthetic Fram+BSO ensemble is smaller (std = 21 TW). While
these indirect estimates show better agreement with OSNAP observations in the latter half of the record, capturing the 2019
minimum and the subsequent increase, they, like the direct transports from the reanalyses, do not reproduce the pronounced
2015 heat-transport peak and, overall, correlate only weakly with OSNAP over the full period (see Table 1). This holds true
across all combinations of datasets and both choke-point approaches, despite the use of multiple, independent data sources.
This discrepancy will be discussed in more detail in section 3.3.

Figure 7 shows (a,b) heat transports along the OSNAP West and East sections, (c,d) the accumulated transports along the
sections and (e,f) the corresponding variability (standard deviation) along the sections. Figure A3 shows the same for volume
transports. Heat transport estimates reproduce the biases found in the velocity and temperature cross-sections (Fig. 4 and 2):
The reanalyses generally agree quite well with observations in terms of the major inflow and outflow branches. However,





**Figure 6.** Time series of heat transports from the OSNAP product, the individual reanalyses and the mean of the four 0.25° reanalyses (GREP) at a) OSNAP West and b) OSNAP East, as well as c) heat transports at OSNAP East estimated via the indirect approach using the GSR and Fram+BSO choke points. All time series are smoothed using a 12-month running mean. Shading for GREP and the indirect estimates represents ±1 standard deviation across the individual products.



| Names | OSNAP | ORAS5 | CGLORS | FOAMv2 | GLORYS2V4 | GLORYS12V1 | GREP | indi.$_{GSR}$ | indi.$_{F+B}$ |
|---|---|---|---|---|---|---|---|---|---|
| East | 417 ± 27 | 369 ± 15 | 398 ± 7 | 402 ± 7 | 412 ± 6 | 402 ± 12 | 395 ± 18 | 370 ± 24 | 374 ± 52 |
| West | 85 ± 6 | 101 ± 21 | 57 ± 7 | 49 ± 11 | 52 ± 13 | 92 ± 12 | 65 ± 26 | - | - |
| Total | 503 ± 28 | 470 ± 12 | 455 ± 10 | 451 ± 13 | 464 ± 15 | 494 ± 12 | 460 ± 28 | - | - |

**Table 2.** Mean heat transports (2014/06–2020/08) at OSNAP East, West, and for the full OSNAP section. Shown are values from the observational product, each reanalysis individually, the mean of the four 0.25° reanalyses (GREP), and the indirect estimates. Uncertainties for individual reanalyses are based on the standard deviation of annual mean values. For GREP and the indirect estimates, uncertainties reflect the spread across the contributing products (calculated as standard deviation).

| Names | ORAS5 | CGLORS | FOAMv2 | GLORYS2V4 | GLORYS12V1 | GREP | indi.$_{GSR}$ | indi.$_{F+B}$ |
|---|---|---|---|---|---|---|---|---|
| OSNAP East | -0.07 | 0.02 | -0.05 | 0.04 | 0.37 | -0.03 | 0.14 | 0.27 |
| OSNAP West | 0.12 | 0.71 | 0.52 | 0.86 | 0.79 | 0.79 | - | - |

**Table 3.** Correlation coefficients of heat transport at OSNAP East and OSNAP West for the period 08/2014–06/2020, comparing observations with ocean reanalyses and (for OSNAP East) indirect estimates.

some discrepancies remain in transport amplitudes and exact positions. At OSNAP West (Fig. 7a,c,e), all reanalyses except ORAS5, overestimate the heat outflow via the LC. A return current slightly east of the LC partially offsets these biases. However, the eastern inflow branch of OSNAP West, associated with the WGC tends to be slightly too weak in all quarter-degree reanalyses, leading to the negative bias in net heat transport at OSNAP West in CGLORS, GLORYS2V4, and FOAMv2.

While GLORYS12V1 also overestimates LC heat transport, this is counterbalanced by stronger heat inflows in the interior of the Labrador Basin. Additionally, heat transport via the WGC, although narrower and stronger than observed, remains in good agreement with observations, resulting in net heat transport estimates close to observed values. In contrast, ORAS5 produces LC heat transports that align well with observations, but stronger heat inflows in the Labrador Basin, combined with slightly underestimated WGC heat transports, lead to a net positive heat transport bias at OSNAP West. At OSNAP East (Fig. 7b,d,f),

heat transports associated with the EGC are slightly displaced in all 0.25° reanalyses and overestimated in GLORYS12V1. Transports in the interior of the Irminger Basin, as well as at the IC and ERRC, are weaker than suggested by the OSNAP observations. In the interior of the Iceland Basin, the reanalyses exhibit high variability and a strong heat inflow and outflow associated with a northward branch of the NAC and a southward recirculation to the east, respectively. This northward and southward circulation feature is not resolved by the OSNAP observations due to OSNAP's array design that relies on end-point

dynamic height moorings to capture the total integrated transport and its variability in the Iceland basin. However, It is worth noting that the accumulated transport over this segment from about 2500 km to 2750 km is consistent between the reanalyses and observations (Fig. 7d). While all reanalyses show positive net heat transport biases when accumulating to the west of Rockall Trough, they tend to underestimate heat transport in the interior of Rockall Trough compared to OSNAP estimates.





**Figure 7.** Mean heat transport across the OSNAP West and East sections for the period 06/2014 to 08/2020. *(a,b)* local heat transport at each point along the section in W per dx with dx=0.25°, *(c,d)* cumulative heat transport along the section, *(e,f)* temporal standard deviation of the local heat transport.

This results in a generally lower net heat transport for the entire OSNAP East section. More broadly, the reanalyses exhibit high variability (Fig. 7c) in regions that lack direct mooring observations, underscoring potential uncertainties in these areas.

### 3.3 Case study: 2015 Heat Transport Anomaly

While differences between observations and reanalyses at OSNAP East persist throughout the OSNAP period (Section 3.2), the most striking mismatch occurs during 2015, when OSNAP shows a pronounced heat transport peak that is absent from




all reanalyses and indirect transport estimates. We use this event as a case study to characterize and localize the differences
between OSNAP and the reanalyses.

Differences in OHT variabilities between observations and reanalyses concerning the 2015 peak can be traced back primarily
to the eastern OSNAP glider region (indicated by the green line in Fig. 1). This region, located in the eastern Iceland Basin
and around Hatton Bank, is highly dynamic and influenced by a northward-flowing branch of the NAC. Figure 8a shows
heat transport anomalies for the glider region alone, where OSNAP observations display a distinct 2015 peak, with transports
approximately 50 to 130 TW higher than the reanalyses, accounting for much of the offset in the full-section transports (Fig.
6b). This peak is also evident in volume transport (Fig. 8b), indicating a link to flow strength or structure rather than temperature
alone. In fact, vertical profiles of temperature anomalies (Fig. 9) show broad agreement across data sets: both indicate colder
than usual conditions in the eastern glider box (blue) and cooler surface waters with warmer intermediate layers in the western
box. However, anomalies tend to be stronger in the reanalyses than in OSNAP. Anomalies of currents (Fig. 9b) reveal a strong
but narrow positive anomaly in OSNAP just west of Hatton bank (at $x \approx 2.85e6$ m) during 2015, suggesting an intensification
of the NAC that is not captured to this extent by any of the reanalyses. The combination of this intensified northward flow and
slightly warmer temperatures relative to the reanalyses likely explains the anomalously high heat transport observed in OSNAP.
Particularly noticeable is also a strong anticyclonic eddy in the western glider region, clearly visible in GLORYS12V1 and,
to a lesser extent, in GLORYS2V4. This feature was also sampled by the OSNAP western glider array, as reported by Lozier
et al. (2017), highlighting the glider system's capability to capture mesoscale variability. However, no such signal is evident in
the publicly available OSNAP dataset used in this study. While this eddy is unrelated to the 2015 heat transport peak, it further
illustrates the variability in this region and the importance of consistent observational coverage.

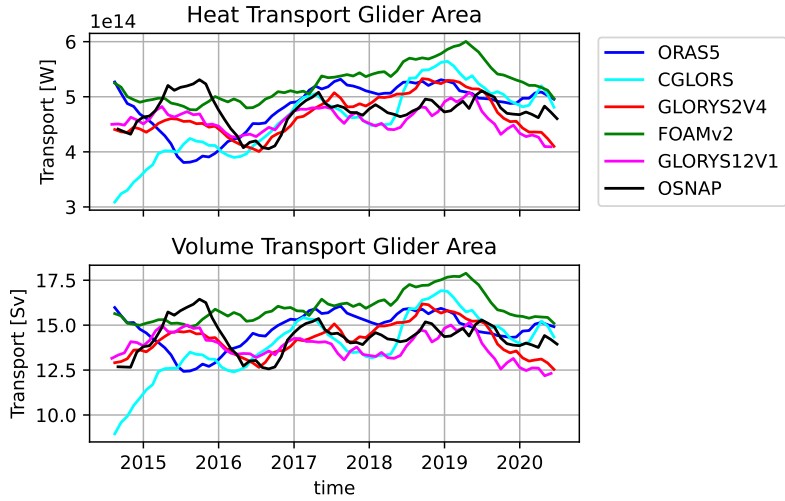

**Figure 8.** Heat (top) and volume (bottom) transports integrated over the eastern glider region, smoothed using a 12-month running mean.
The definition of the glider region can be seen in Fig. 1 and in Fig. 9.



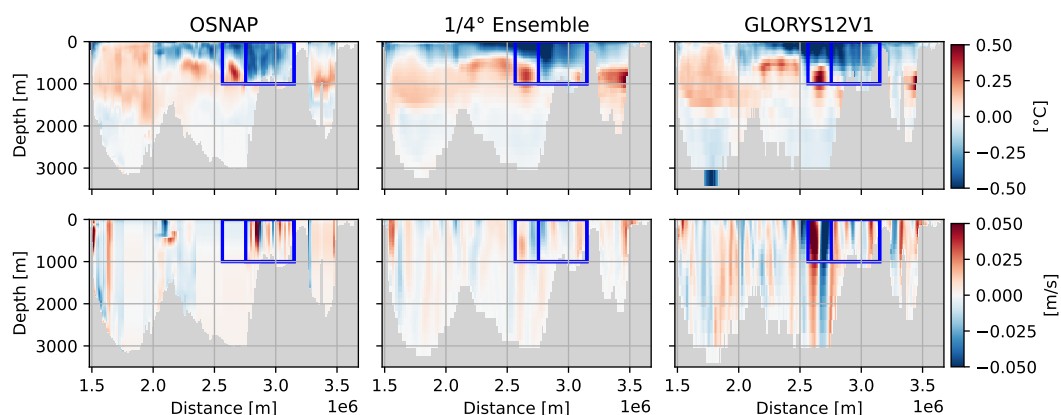

**Figure 9.** Temperature (top) and velocity (bottom) anomalies for the 2015 period for OSNAP (left) the mean of all four quarter degree reanalyses (middle) and GLORYS12V1 (right). The eastern and western glider region are indicated by the blue box.

To provide geostrophic context for the 2015 anomaly in the glider region, we next examine along-section sea-level anomalies (SLA) and their gradients. We note a key methodological difference: OSNAP derives time-varying geostrophic shear from T/S
and applies a time-mean reference velocity with a uniform barotropic compensation, whereas the reanalyses assimilate SLA and thus have time-varying surface geostrophic flow tied to SLA. Accordingly, the SLA–transport comparison below is intended as a diagnostic of reanalysis consistency and as context for OSNAP. In this framework, tight pointwise correlations between SLA gradients and OSNAP transport are not expected. Nevertheless, because along-section SLA gradients set the surface geostrophic shear, we'd expect a coherent relationship after spatial averaging over broader segments (e.g., the glider regions).
Figure 10 presents Hovmöller diagrams of SLA (left), its along-section gradient (middle), and top-to-bottom vertically integrated volume transport (right), based on observations (top row), GLORYS12V1 (middle), and GLORYS2V4 (bottom). SLA in the reanalyses is computed relative to the same 1993–2012 reference period as the observational product. Overall, SLA and its gradients show similar spatial and temporal patterns between the observations and the reanalyses, consistent with the fact that both GLORYS products assimilate SLA data even at these northern latitudes. Fine-scale details are more apparent in
the higher-resolution observations and GLORYS12V1 than in the coarser GLORYS2V4.

However, the vertically integrated volume transport fields reveal more pronounced differences. In OSNAP, the strong 2015/16 transport maximum is not accompanied by an equally strong along-section SLA gradient in the eastern glider region, and outside this event the OSNAP transports remain relatively smooth and weak, even during periods of enhanced SLA gradients. This is consistent with OSNAP's use of a time-mean reference velocity with barotropic compensation, i.e., SLA does
not directly set the absolute velocity along the line. The GLORYS reanalyses, which assimilate SLA, exhibit spatially coherent transport anomalies, including pronounced signals in the western glider box, that align more closely with their SLA-gradient fields.



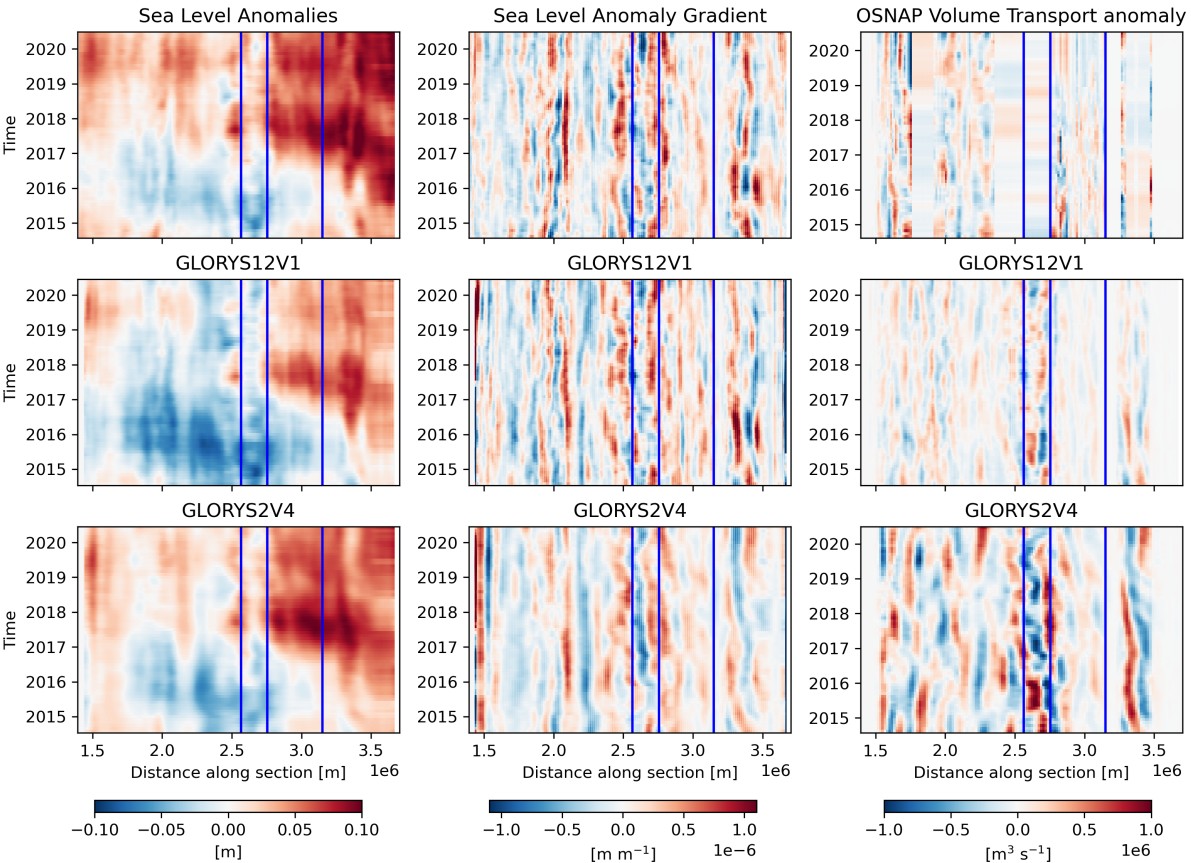

**Figure 10.** *left:* Sea Level anomalies along the OSNAP East section referenced to the 1993-2012 period from satellite observations, GLO-RYS12V1 and GLORYS2V4, *middle:* the respective Sea Level anomaly gradients, *right:* vertically integrated volume transports for OSNAP, GLORYS12V1 and GLORYS2V4. Blue lines indicate the glider areas.

Correlations between SLA gradients and volume transport across the glider region are shown in Fig.11. Solid lines represent correlations between each reanalysis and its own SLA gradient field, while dashed lines represent correlations to the observational SLA gradient. As expected given OSNAP's time-mean reference velocity, OSNAP shows weak pointwise correlations with observed SLA gradients (mean ≈ 0.08 along the section). In contrast, GLORYS12V1 and GLORYS2V4 exhibit higher correlations with observed SLA gradients (0.40 and 0.39, respectively) and even larger values when compared to their own SLA fields (0.79 and 0.51). Notably, the higher-resolution GLORYS12V1 shows the strongest correlations overall, consistent with its improved spatial representation of circulation features. Correlations are weaker for the ORAS5 reanalysis (not shown), with a correlation of just 0.25 against observed SLA gradients, likely a result of it not assimilating sea level anomalies north of 50°N. To reduce noise and better match the spatial scales at which geostrophic balance is maintained, all data were smoothed to 1° resolution. This allows us to focus on mesoscale dynamics while suppressing small-scale variability and potential sam-



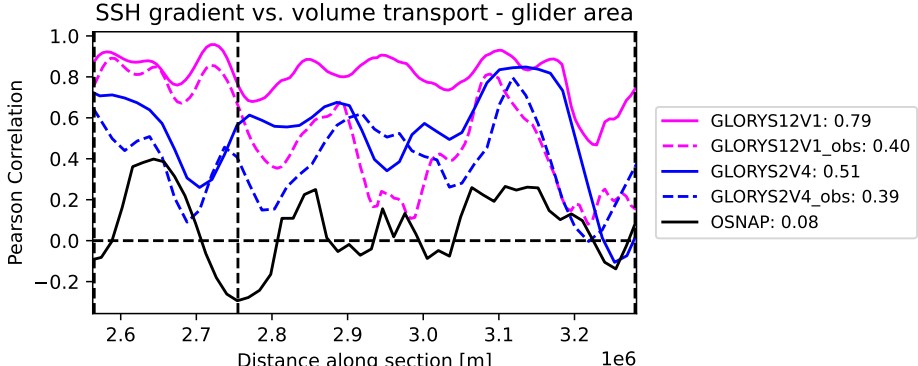

**Figure 11.** Along section temporal correlations between the SLA gradient and full-depth integrated volume transports for the glider region, smoothed to 1° resolution. Solid lines show correlations between each reanalysis and its own SLA gradient, dashed lines show correlations to the observational SLA gradient. Values in the legend describe correlations averaged over the whole glider region.

pling mismatches. Despite this smoothing, the pointwise correlations across the section remain relatively low for the OSNAP dataset. However, when SLA gradients and volume transport are averaged over the two full glider regions before computing

the correlation, substantially higher values are found for OSNAP (0.55 for the western, 0.47 for the eastern glider region) and GLORYS12V1 (0.72 for the western, 0.89 for the eastern), while the correlation for GLORYS2V4 decreases slightly (0.32 for the western, 0.37 for the eastern). These reanalysis-based values refer to correlations with their respective SLA gradient fields. Overall, these results suggest that, while local correlations are sensitive to methodology, sampling, and the limited ability of observations and reanalyses to resolve small-scale processes, the large-scale geostrophic relationship emerges clearly after

spatial averaging over the glider regions in both the observations and the high-resolution reanalyses.

Correlations computed using full geostrophic velocity fields, calculated from reanalysis temperature and salinity profiles via pressure gradients and referenced to surface altimetry show nearly identical patterns (not shown), confirming that SLA gradients dominate the transport signal.

## 3.4   AMOC

To complement the heat transport analysis, we also assess the Meridional Overturning Circulation (AMOC) in density space at OSNAP East (Fig. 12). The reanalyses reproduce the mean overturning streamfunction reasonably well, consistent with earlier findings by Baker et al. (2022), with a maximum around $27.5$–$27.6 \, \mathrm{kg m^{-3}}$ and peak strengths of about 16 Sv, slightly higher in the reanalyses, particularly in ORAS5. In terms of variability, OSNAP observations show a clear peak in 2015, which coincides with the observed heat transport variability (see Fig. 6). Interestingly, the reanalyses also display a peak in overturning strength,

though shifted slightly later, to early 2016. However, and as discussed in section 3.2, this overturning peak does not coincide with a heat transport peak in the reanalyses. Looking at transports of the in- and outflow branches of the overturning, we find that in GLORYS12V1 both inflow and outflow heat transports are stronger than in OSNAP, but the outflow has a greater net




effect than in OSNAP. This suppresses the overall heat transport peak despite intensified overturning. The mismatch may result from temperature biases, such as colder inflow or warmer outflow, and suggests that the overturning in the reanalyses may involve different water mass properties or pathways than observed.

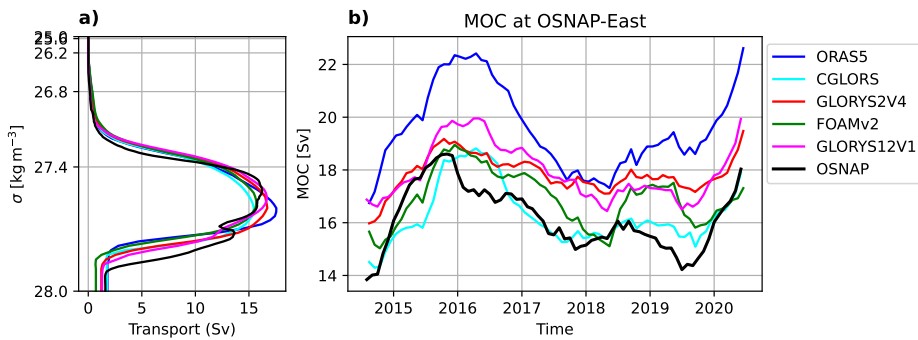

**Figure 12.** a) Meridional Overturning Circulation (MOC) streamfunction in density space, averaged over the period 06/2014 to 08/2020. b) Time series of the MOC at the $\sigma_0$ level of maximum overturning, smoothed using a 12-month running mean.

To diagnose why the reanalysis MOC peak does not translate into a total OHT peak, we decompose MHT into overturning and gyre components (Fig. 13), as done e.g. by Li et al. (2021). In the OSNAP observations, the 2015 maximum is overturning-driven and followed by a multi-year decline. The 2019 minimum is likewise set by the overturning term and the consecutive increase is reinforced by a strong gyre contribution. In the reanalyses, the overturning term is biased high in the mean but its 2015 anomaly is weaker and lacks the subsequent decline, while the gyre term is systematically lower than in OSNAP (even negative in ORAS5). The 2019 minimum in the overturning term is weaker in the reanalyses and a contrasting maximum in the gyre term flattens the 2019 minimum in the net transports (see Fig. 6). Therefore, the good correspondence in MOC yet poor correspondence in total OHT might be due to the temperature contrast associated with the overturning being too low or shifted in the reanalyses and additionally the gyre term dampening the total OHT anomalies.

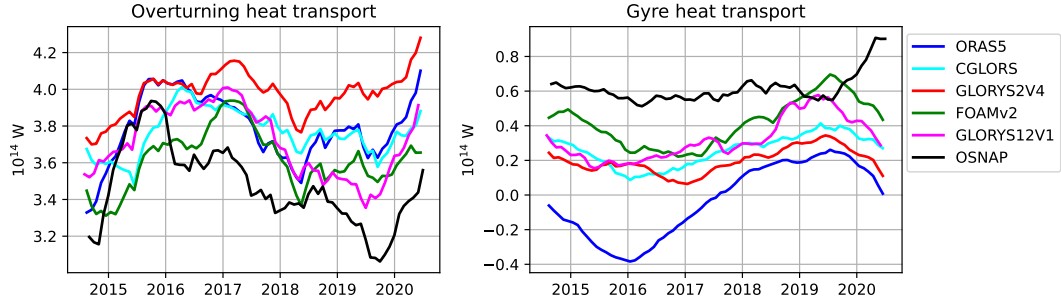

**Figure 13.** Meridional heat transport decomposition in density space at OSNAP-East for the 08/2014 to 06/2020 period, smoothed with a 12-month running mean: (a) overturning component; (b) gyre component.



## 4 Conclusions

This study evaluates the ability of global ocean reanalyses to capture oceanic heat transport across the OSNAP section by comparing reanalysis-derived estimates with observational estimates of heat transport, velocity, and temperature cross-sections. While ocean reanalyses generally reproduce the broad structure of the AMOC and its associated heat transport, systematic biases persist in both the thermal and dynamic properties of the ocean circulation.

Reanalyses successfully capture the major inflow and outflow branches at OSNAP, but discrepancies remain in transport amplitudes and current positioning. At OSNAP West, most reanalyses overestimate the heat transport via the LC, while underestimating the inflow associated with the WGC, leading to a net negative heat transport bias. The higher-resolution GLORYS12V1 better represents heat transport through the WGC and interior Labrador Basin, resulting in values closer to observations. Conversely, ORAS5, while producing more realistic LC transports, shows an overall positive heat transport bias due to excessive heat inflow within the Labrador Basin.

At OSNAP East, reanalyses show eastward displacements of the EGC and weaker transports in the Irminger Basin, Iceland Basin, and at the ERRC. A significant heat transport inflow and outflow associated with NAC branches is present in the Iceland Basin (Fig. 7b), which is not captured by the OSNAP observations due to sparse mooring coverage in this region. This discrepancy highlights the impact of observational gaps on our understanding of the spatial structure of meridional heat transport. However, we note that the integrated transports over this area are generally consistent between observations and reanalyses (Fig. 7d).

The temporal variability of heat transport is generally well represented at OSNAP West, with high correlations between reanalyses and OSNAP observations. However, at OSNAP East reanalyses disagree with the observed variability throughout the period, showing muted amplitudes and missing the 2015 maximum and the 2019 minimum. Decomposing MHT into overturning and gyre contributions shows that in the observations variability is overturning dominated, whereas in the reanalyses the overturning term is biased high in the mean but the 2015/2019 anomalies are weaker, and additionally flattened by the gyre term. A possible reason why MOC agreement in 2015 does not translate into OHT agreement is that the temperature contrast within the overturning limb is reduced or shifted to other regions in the reanalyses. While the event may be genuine and missed by the reanalyses, its absence from independent budget-based estimates and the lack of a clear sea-level signature argue for caution. This highlights both the value of gliders in resolving fine-scale circulation and the need for sustained, multi-platform observational strategies and cross-validation with models and indirect methods to robustly assess heat transport variability in dynamically active regions.

These findings emphasize both the value and limitations of ocean reanalyses in representing heat transport. While they provide a useful tool for temporally and spatially extending heat transport estimates beyond direct observations, persistent biases highlight the need for continued improvements in data assimilation techniques, model resolution, and observational coverage. In particular, the lack of direct measurements in dynamically complex regions such as the Iceland Basin and Rockall Trough, the deep ocean and basin interiors complicates the assessment of uncertainties in both reanalyses and OSNAP observations.





While well-maintained mooring lines provide the gold standard for MHT variability estimation, it is remarkable that discrepancies to reanalyses can be used to trace sampling issues and inhomogeneities in those observing systems. There are many

examples in atmospheric sciences where reanalyses could be used to find and even estimate biases in global observing systems (Hollingsworth et al., 1986; Haimberger et al., 2012). We therefore see this as an indication of the high value and quality of present ocean reanalyses.

Nevertheless, addressing data scarcity, particularly in the deep ocean and overflow regions, is essential not only for improving the accuracy of reanalyses and reducing uncertainties in reconstructed oceanic heat transport but also for strengthening direct

observational OSNAP heat transport estimates. Expanding sustained and spatially comprehensive ocean observations will provide critical constraints for both data assimilation and observational estimates, reducing biases and uncertainties in AMOC variability assessments. In this regard, coordinated international efforts such as the Marine Environment Reanalyses Evaluation Project (MER-EP; UNESCO Ocean Decade, 2025) are essential for advancing the systematic evaluation of ocean reanalyses and for ensuring their suitability as reliable tools for climate research.

**Appendix: Supplemental figures**

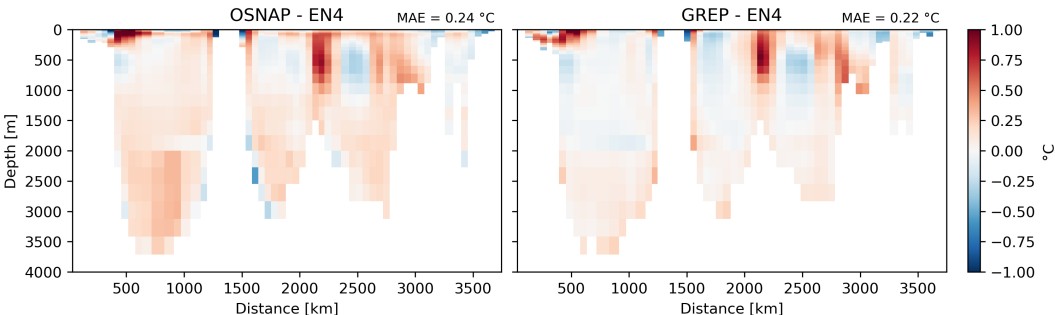

**Figure A1.** Temperature differences between OSNAP observations and the EN4 objective analysis (left) as well as the GREP reanalyse mean and EN4 (right), averaged over the period 08/2014 to 06/2020. Positive values indicate regions where OSNAP/GREP is warmer than EN4.



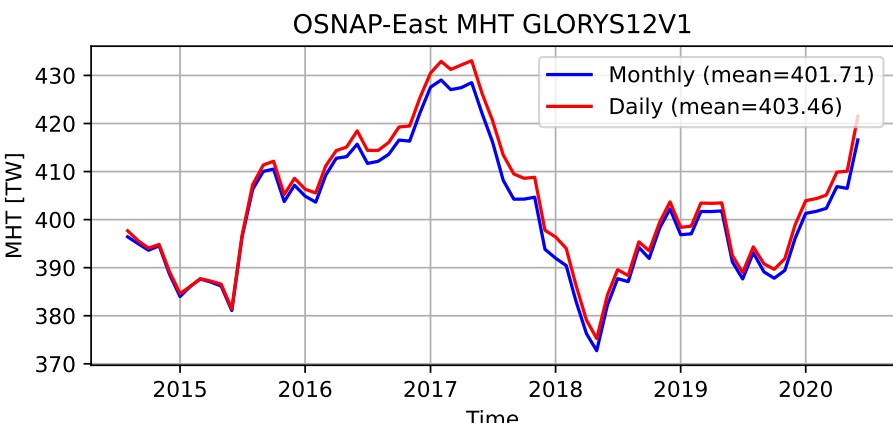

**Figure A2.** Time series of MHT at OSNAP East derived from monthly and from daily GLORYS12V1 data, smoothed using a 12-month running mean.





**Figure A3.** Mean volume transport across the OSNAP West and East sections for the period 06/2014 to 08/2020. *(a,b)* local volume transport at each point along the section in m$^3$/s per dx with dx=0.25°, *(c,d)* cumulative volume transport along the section, *(e,f)* temporal standard deviation of the local volume transport.



*Author contributions.* SW, MM and LH conceptualized the study. SW and IW performed the data analysis, including the production of the figures. SW prepared the manuscript. MM, LH and YF contributed to the interpretation of results and the writing of the manuscript. All authors have read and agreed to the publication of the present version of the manuscript.

*Acknowledgements.* This work was funded by the Copernicus Marine Environment Monitoring Service under contracts 21003-COP-GLORAN Lot 7 and 24254L03-COP-GLORAN MER-EP 4000, as well as by the European Space Agency (ESA) under contract 4000145298/24/I-LR (MOTECUSOMA) and contract 4000147586/I/25-LR (CCI Ocean Surface Heat Fluxes). We would like to thank Yuying Pan for the helpful discussion concerning the indirect heat transport calculations. ChatGPT (OpenAI) and DeepL were used to improve grammar and phrasing.

*Code availability.* Codes for calculating the figures will be made available on https://zid.univie.ac.at/en/gitlab/

Straitflux is available via https://pypi.org/project/StraitFlux/

*Data availability.* Source data have all been taken from public sources as cited. Those have not been copied to a public repository. Input data to reproduce the plots of this manuscript will be provided at https://zid.univie.ac.at/en/phaidra/



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
