# Peer review of "Subpolar Atlantic meridional heat transports from OSNAP and ocean reanalyses - a comparison"

_EGUsphere, 2025_

## Author Comment (AC1)

Thank you very much for your positive comments and constructive feedback, you addressed some important points. Your clarifications helped to make the manuscript clearer for the reader. Our responses are provided in green (changes made in the manuscript are written in **bold**) together with your original comments in black.

We really appreciate your time and insight in reviewing our manuscript!

Kind regards,
Susanna (on behalf of all co-authors)

**Reviewer #1:**

Overarching comment:

Heat transport cannot be calculated for cross-sections that do not conserve mass because the results are sensitive to the choice or reference temperature. Either the authors need to show that the lines are mass-conserving (or at least volume-conserving), or use the term "temperature transport" rather than "heat transport" (as in Johns et al. 2011). To this end, it would be good to show the net volume transport for each of the reanalyses. This is shown to some degree in Fig. A3, but a time series of volume transport each month for each product would be enlightening.

We agree that strictly unambiguous heat transports would require closed (or at least volume-conserving) sections and that otherwise the transport depends on the choice of reference temperature. As discussed by Schauer and Beszczynska-Möller (2009) and related studies, that condition is generally not fulfilled for partial sections such as OSNAP East and West.

In line with common practice in the literature (e.g. Tsubouchi et al., 2012, 2018; Muilwijk et al., 2018; Shu et al., 2022; Heuzé et al., 2023), we therefore compute heat transports relative to a fixed reference temperature ($\theta_{ref} = 0$ °C).
We use the term "heat transport" in this conventional, reference-dependent sense rather than "temperature transport", as the transported quantity **represents the enthalpy of seawater rather than temperature itself**, and to maintain consistency with previous studies (Winkelbauer et al., 2024a; Winkelbauer et al., 2024b).

We state this more clearly in Section 2.1 now (L128 and following):
*Additionally, the potential temperature $\theta$ and a reference temperature $\theta_{ref}$ are needed for estimating heat transports.* ***Strictly speaking, unambiguous heat transports would require closed mass transports across the examined section, which is generally not the case for partial sections such as OSNAP East and West and only approximately satisfied for the total oceanic transport (Schauer and Beszczynska-Möller, 2009). Therefore, heat transports depend on the choice of reference temperature. To minimize the ambiguity arising from this choice, $\theta_{ref}$ should be chosen to represent the section-mean temperature of the flow across the considered section (e.g. Bacon et al., 2015). However, to ensure internal consistency across products***

*we calculate all heat transports relative to a constant reference temperature of θ_ref = 0 °C, following common practice (e.g., Tsubouchi et al., 2012, Muilwijk et al., 2018; Shu et al., 2022; Heuzé et al., 2023). Throughout this study, the term "heat transport" is therefore used in this conventional, reference-dependent sense and describes the transport of heat referenced to water at 0°C.*

To further clarify the degree of volume imbalance, we now include time series of net volume transport for each reanalysis product in the Appendix:

[Figure]

**Figure A3. Net volume transport across the eastern and western OSNAP section derived from OSNAP observations (computed from the gridded sections) and for each reanalysis.**

We agree that the seen differences in volume transport variability can in principle affect reference-dependent heat transports. To quantify the possible impact of this effect we perform a back-of-the-envelope estimate based on the full volume-transport time series:

$$\Delta OHT_{max}(t) = \rho c_p \Delta V(t) \Delta \theta(t)$$

where $\Delta V(t)$ is the net volume transport across the section and $\Delta \theta$ is the mean temperature bias between OSNAP and the respective reanalysis.

The resulting time series of $\Delta \text{OHT}_{max}$ is shown in the figure (right Plot) below. The mean value across all reanalyses is approximately **1.2 TW**, more than two orders of magnitude smaller than the mean heat transport (~400 TW) and more than an order of magnitude smaller than the observed 2015 anomaly (~60 TW).

This demonstrates that net volume-transport imbalances and reference-temperature ambiguity cannot explain the 2015 discrepancy, which must instead be dominated by differences in the thermal structure and spatial distribution of the flow.

[Figure]

Fig.: *left*: mean OHT at OSNAP East; *right*: uncertainty in OHT due to OVT imbalances estimated via a back-of-envelope approach

We added the following paragraph to the manuscript:

***Volume transport time series are shown in Fig. A1. OSNAP-derived net volume transport (computed from the gridded sections) across OSNAP East has a mean comparable to the reanalyses but shows substantially reduced variability. This is expected because OSNAP's velocity reconstruction combines time-varying geostrophic shear with a constrained barotropic component (transport closure/compensation), which dampens section integrated volume-transport variability. In contrast, ocean reanalysis permit time-varying transports in response to atmospheric forcing and freshwater fluxes, leading to larger variability in net volume transport across the open OSNAP section. While the realism of this variability cannot be independently assessed here, its magnitude remains small compared to what would be required to explain the pronounced heat-transport anomaly in 2015. A conservative upper-bound estimate shows that the associated reference-temperature-dependent contribution to heat transport at OSNAP East is of order 1–2 TW on average, which is negligible compared to the mean transport (~400 TW), indicating that differences in the thermal structure and distribution of the flow play a more dominant role in the 2015 anomaly.***

Specific comments:

l. 54-55: "As reanalyses generally do not assimilate direct observations of ocean currents, their transport estimates depend largely on model dynamics and parameterizations rather than observational constraints" – this is not entirely true. Ocean reanalyses assimilate SSH and T/S, which together constrain the geostrophic circulation. Most of the AMOC (and resulting MHT) is in geostrophic balance, thus the components of the velocity field that are important to this paper are indeed assimilated. The one exception to this would be the boundary currents, where direct velocity measurements from ADCPs and current meters are indeed not assimilated by the reanalyses. This sentence should be rewritten to convey this information.

We thank the reviewer for this clarification and agree that the original wording was imprecise. While ocean reanalyses do not assimilate direct current measurements, they do assimilate sea level anomalies and temperature/salinity profiles, which together strongly constrain the large-scale geostrophic circulation. We have revised the text to reflect this:

*As reanalyses generally do not assimilate direct observations of ocean currents, their transport estimates depend on a combination of model dynamics, parametrizations **and observational constraints provided indirectly through the assimilation of sea level anomalies and temperature/salinity profiles. Since much of the heat transport associated with the AMOC is in geostrophic balance, these components of the velocity field are indirectly constrained by observations, while limitations remain particularly for boundary currents and narrow passages, where direct velocity measurements are not assimilated.***

l. 83-85: are the vertical cross-sections from GLORYS12V1 re-mapped onto a ¼° grid to be comparable to the other reanalyses? If not, the mean RMSE shown in Figs. 3 and 5 could be aliased by the different spatial resolution.

The GLORYS12V1 sections were not remapped first to a ¼° grid. Instead, all reanalyses are interpolated directly onto the same OSNAP gridded section prior to the calculation of biases and RMSE. This effectively evaluates all products at the OSNAP resolution and smooths higher-resolution features in GLORYS12V1. As the OSNAP grid is similar coarse as the ¼° grid we do not expect aliasing problems. Nevertheless, to assess whether the different native resolutions could bias the RMSE, we additionally tested to remap GLORYS12V1 first to the ¼° grid and then interpolated onto the OSNAP grid. The resulting RMSE values are virtually unchanged, suggesting that the differences in native model resolution do not alias the RMSE values shown in Figs. 3 and 5.

l. 86: "...they differ in their data assimilation methods..." it would be good to clarify what these differences are. A table would be a good way to organize this information.

We added the following Table in section 2.1

| Reanalysis | Resolution | Assimilated data | Assimilation scheme |
|---|---|---|---|
| ORAS5 (Zuo et al., 2019) | 1/4° | T/S profiles, SLA (50°S-50°N), SST, SIC | NEMOVAR (3D-Var) (Mogensen and Balmaseda, 2012) |
| CGLORSv7 (Storto and Masina, 2016) | 1/4° | T/S profiles, SLA (global, ice-free), SST, SIC | OceanVar (3D-Var) (Dobricic and Pinardi, 2008) |
| GLORYS2V4 (Garric and L.Parent, 2016) | 1/4° | T/S profiles, SLA (global, ice-free), SST, SIC | SAM2 (SEEK, multivariate) |
| FOAMv2/GloRanV14 (MacLachlan et al., 2015) | 1/4° | T/S profiles, SLA (global, ice-free), SST, SIC | NEMOVAR (3D-Var) |
| GLORYS12V1 (Lellouche et al., 2018) | 1/12° | T/S profiles, SLA (global, ice-free), SST, SIC | SAM (SEEK, multivariate) |

**Table 1.** List of used ocean reanalyses. All reanalyses assimilate temperature and salinity profiles and sea level anomalies, which constrain the large-scale geostrophic circulation. None of the products assimilate direct velocity observations from current meters or ADCPs, including those from the OSNAP array.

l. 90: "they can be considered independent of OSNAP in that regard". As mentioned above, though the velocities are not assimilated, much of the OSNAP velocity field is determined from SLA and geostrophy so the only place there is any independence is in the boundary currents. This should be specified.

We rephrased that part to:

*All reanalyses assimilate in situ temperature and salinity profiles and SLA, which constrain the large-scale geostrophic circulation,* ***but none assimilate direct velocity observations from current meters or ADCPs. In contrast, OSNAP transport estimates are derived from direct, full-depth observations of velocity, temperature, and salinity obtained from moorings, gliders and hydrographic measurements. As a result, OSNAP and the reanalyses differ fundamentally in how ocean velocities, in particular boundary currents, are constrained.***

Fig. 1: what is the mooring in the center of the Labrador Sea?

We removed that mooring from the figure as it is not used in the OSNAP calculation.

l. 133: the reanalyses used in this paper are not volume (or mass) conserving so to which 'conservation properties' are the authors referring?

We clarify that the ocean reanalyses used in this study are indeed volume conserving, even though the net volume transport across an open section such as OSNAP East or West is not required to be zero at monthly timescales.

By "conservation properties" we refer to the numerical consistency of fluxes as represented on the native model grids. Interpolating velocity fields prior to transport

calculations can introduce spurious signals and compromise the model-internal conservation of fluxes. We have clarified this point in the revised manuscript:

*To avoid interpolation **and preserve the numerical consistency of fluxes on the native model grids**, net integrated transports from reanalyses are calculated using StraitFlux's line-integration method.*

l. 136: when the heat transports are calculated at monthly time scales, is it calculated from the monthly mean of the heat transport or calculated from the monthly mean velocity and temperature fields? The former accounts for the v'T' term, while the other does not.

In general, heat transports in this study are calculated from monthly mean velocity and temperature fields (for data availability reasons) and therefore do not explicitly include sub-monthly covariance terms (v'T' term). However, the comparison to daily GLORYS12V1 data discussed at line 136 is based on daily vT products and therefore does include the v'T' term. As discussed, the resulting differences between transports computed from daily fields and those derived from monthly mean fields are comparatively small, with negligible impact on temporal variability. This demonstrates that neglecting sub-monthly covariance terms does not substantially affect the results presented in this study. We adapted the wording slightly:

*All transport calculations in this study are based on monthly mean output from the ocean reanalyses. To evaluate the potential influence of temporal resolution, we additionally tested calculations based on daily velocity and temperature fields for GLORYS12V1, **which include sub-monthly covariance terms (v'T') that are not explicitly resolved when using monthly mean fields. The resulting differences in integrated heat transport across the OSNAP section amount to about 2 TW on average over the analysis period (corresponding to approximately 0.5% of the mean transport, see Fig. A2), with negligible impact on variability. This indicates that monthly output provides a sufficiently accurate representation for the purposes of this study.***

l. 136: the authors refer to a 0.5% error... is this a percentage of PW? Heat transport has very small variability compared to its mean value. So it would be more clear if the authors just reported a value of heat transport in PW rather than a %.

We have revised the text to report the difference in heat transport in TW, with the percentage given only for reference. See comment above.

l. 157: What is meant by "Mass-consistent heat transport estimates"?

By "mass-consistent heat transport estimates" we refer to heat transports inferred from atmospheric energy budgets that are explicitly constrained to satisfy mass continuity. We have clarified this wording in the revised manuscript:

*Heat transport estimates inferred from mass-consistent atmospheric energy budgets (see, e.g., Mayer et al., 2021, Mayer et al., 2024) are used at two different choke-points: the Greenland–Scotland Ridge (GSR), and the combination of Fram Strait (FS) and the Barents Sea Opening (BSO).*

Table 1: this is an impressive list of data sets. Why was JRA-55 used rather than the updated version (JRA-3Q)?

We have already started producing mass consistent inferred surface heat fluxes based on the newer JRA-3Q reanalysis. However, these JRA-3Q energy budgets are still under consolidation and have not yet been formally published or fully documented. Therefore, we rely on the JRA55-based budgets in this work.

We added the following to the manuscript:

*Fs is estimated indirectly from atmospheric budgets, so these are much better constrained by independent observations than parameterized surface fluxes, which typically are more uncertain and depend on the sea state (Mayer et al., 2023; Trenberth et al., 2019). Therefore, divergences and tendencies from atmospheric reanalyses ERA5 (mass-consistent energy budgets, Mayer et al., 2021a), MERRA2 (Gelaro et al., 2017) and JRA55 (Kobayashi et al., 2015) are combined with top-of-atmosphere (TOA) fluxes from CERES-EBAF TOA version 4.2 (Scott et al., 2022; NASA/LARC/SD/ASDC, 2025). An updated implementation based on the newer JRA-3Q (Kosaka et al., 2024) reanalysis is currently under development and will be addressed in future work once the corresponding energy budgets are fully consolidated.*

Fig. 3: Consider using a different colorbar to depict RMSE – at first look, this appears as a consistent high bias in the reanalyses compared to OSNAP.

We have revised Fig. 3 and Fig. 5 to use a more neutral colormap for RMSE, which more clearly represents error magnitude without implying a systematic bias.

l. 315 and 404: it is unclear to me whether this 2015 event was captured by OSNAP because there was a glider in that year (and not afterwards), or if this was truly an anomalous event. It would be interesting to analyze an OSNAP gridded section that does not include the glider in 2014-2016. Does the event appear if the glider is not included? Determining whether the event is real or an artifact of changing observational structure would go a long way toward understanding the authors' thoughts in the conclusions about the importance of a consistent set of observations.

We thank the reviewer for raising this important point. We first clarify that the discussion at line 315 refers to a mesoscale eddy observed in the western glider region, which we explicitly state is unrelated to the 2015 heat transport peak. The 2015 anomaly discussed at line 404, by contrast, refers to the basin-scale heat transport maximum at OSNAP East, which we trace primarily to an intensified NAC inflow in the eastern glider region.

We agree that assessing the sensitivity of the OSNAP heat transport estimates to changes in the observing system would be valuable. However, recomputing OSNAP transports without glider data is beyond the scope of this study and not feasible with the publicly available OSNAP gridded product.
To acknowledge this uncertainty, we have added a brief statement to the Discussion:

***We cannot exclude that changes in observational coverage, including the use of gliders early in the OSNAP record, may contribute to the observed amplitude of the 2015 heat transport peak. Future sensitivity studies assessing OSNAP transport estimates with and without specific observing components, such as gliders, could help further quantify the impact of observational heterogeneity.***

Fig. 11: the units on the x-axis are a bit strange... 2.5-3.3 x $10^6$ m... I suggest using km and specifying that this refers to the along-section distance from the OSNAP western boundary.

We changed the units to km.

l. 412-418: In this paragraph, the authors express more confidence in reanalyses products than is justified from the results of this paper. While it is true that the discrepancy in OHT between OSNAP and the reanalyses in 2015 is interesting and raises questions about the coverage and consistency of OSNAP, the authors have not presented any independent evidence that reanalyses can provide error estimates for the observing system (OSNAP in this case). I agree that this is a possible use of ocean reanalyses once they are validated, but the authors would need to present independent data that justify this usage. Given how much the reanalyses disagree with one another (in this paper and in others, e.g. Jackson et al. 2019), I would proceed down this path with a lot of caution – and much more caution than interpreting the direct observations from OSNAP.

We agree that our original wording overstated the degree of confidence that can currently be placed in ocean reanalyses as tools to diagnose errors in observational systems. Our intention was not to suggest that reanalyses can provide quantitative error estimates for OSNAP, but rather that systematic and coherent discrepancies across multiple reanalyses may help highlight regions where both observing systems and models are challenged. We have revised the paragraph to:

*While well-maintained mooring lines provide the gold standard for MHT variability estimation, systematic discrepancies between OSNAP and reanalyses **may be used to find regions of increased uncertainty arising from limitations in both observing systems and models.** There are many examples in atmospheric sciences where reanalyses could be used to find and even estimate biases in global observing systems (Hollingsworth et al., 1986; Haimberger et al., 2012). **In this sense, our results suggest that present ocean reanalyses can serve as a valuable complementary tool for diagnosing uncertainty.***

Conclusions: the authors could also mention the use of reanalyses to replace the use of moorings in regions where lower frequency variability is dominant. This would save costs and is currently being pursued by the RAPID team (Petit et al., (in review)).

We added a paragraph in the conclusion section:

*While well-maintained mooring lines provide the gold standard for MHT variability estimation, systematic discrepancies between OSNAP and reanalyses may be used to find regions of heightened uncertainty arising from limitations in both observing systems and models. There are many examples in atmospheric sciences where reanalyses could be used to find and even estimate biases in global observing systems (Hollingsworth et al., 1986; Haimberger et al., 2012). In this sense, our results suggest that present ocean reanalyses can serve as a valuable complementary tool for diagnosing uncertainty.*
***At the same time, in regions where low-frequency variability dominates and where reanalyses demonstrate robust skill, reanalysis products may offer complementary means to extend or support observational estimates (see e.g., Mayer et al., 2023, Fritz et al., 2023). Such approaches are currently being explored within the RAPID program as part of efforts to develop more sustainable and cost-effective long-term observing strategies (Petit et al., 2025).***

**References:**

Johns et al. (2011): https://doi.org/10.1175/2010JCLI3997.1

Jackson et al. (2019): https://doi.org/10.1029/2019JC015210

---

## Author Comment (AC3)

Thank you very much for your positive comments and constructive feedback, you addressed some important points. Your clarifications helped to make the manuscript clearer for the reader. Our responses are provided in green (changes made in the manuscript are written in **bold**) together with your original comments in black.

We really appreciate your time and insight in reviewing our manuscript!

Kind regards,
Susanna (on behalf of all co-authors)

**Reviewer #2**

**Major Comments**

1. As I understand it, the authors use the OSNAP data as gold standard for the evaluation of OHT in different reanalyses. However, they repeatedly conclude that most of the differences could be attributed to a lack of observations at specific depths and areas along the section, as OSNAP uses constant fields in the interior and relies on end-point dynamic height moorings to capture the total integrated transport and its variability (e.g. of conclusion at l.213, l.229, l.233, l.289, l.295, l.316, l.394, l.411). In this context, can the authors comment on the choice of OSNAP data as a gold standard to evaluate these reanalyses while there are not enough OSNAP observations in the interior to conclude anything on the possible causes for the total OHT differences? In a way, the reanalyses might assimilate more observations in the interior than the OSNAP product, meaning that their OHT distribution could be considered as closer to the truth than the one from the OSNAP product. Hence, how can we reliably assess inconsistencies between reanalyses and OSNAP?

   We appreciate this thoughtful comment. We would like to clarify that we do not intend to use OSNAP as a gold standard in the sense of providing a definitive "truth". Rather, it represents the most comprehensive direct observational estimate of heat transport across the subpolar North Atlantic currently available. Our analysis is therefore framed as a **comparison** (not a validation) between two fundamentally different approaches: direct observations with incomplete spatial coverage, and reanalyses that assimilate a range of datasets but rely on imperfect model dynamics and parameterizations. The purpose is not primarily to determine which product is closer to the truth, but to identify where and why these approaches diverge, thereby highlighting regions and processes that remain uncertain.

2. A specific example of my main comment #1 is the discussion of an anticyclonic eddy in the NAC region at lines 309–317. Can the authors clarify their conclusion: is the inconsistency coming from the resolution of the reanalyses at the boundaries or from a lack of observations in the available OSNAP dataset? If

these two points are valid, how a validation of one dataset as compared to the other can be convincing? Related to this specific point, I am not sure to understand why the anticyclonic eddy can be observed in the glider data (as discussed in Lozier et al., 2017) but not in the final OSNAP product?

As clarified in our response to Comment #1, we do not use one product to validate the other. Rather, our analysis is framed as a comparison between two complementary approaches in order to identify where and why they diverge.

We emphasize that the paragraph at lines 309–317 refers to two distinct circulation features that serve different purposes in the discussion:

Lines 301–312 refer to a narrow, intensified northward current anomaly near Hatton Bank in 2015, observed in the **eastern** OSNAP glider region. This feature is directly linked to the 2015 heat transport peak and reflects a localized strengthening of the NAC that is more pronounced in the OSNAP observations than in any of the reanalyses.
**The source of the inconsistency is difficult to attribute uniquely to either reanalysis resolution or OSNAP sampling with the information available.**

Lines 313–317, by contrast, describe a separate anticyclonic eddy in the **western** glider region, which is explicitly stated to be unrelated to the 2015 heat transport peak. This second example is included only to illustrate the strong mesoscale variability in the glider regions and their sensitivity to both (i) model resolution/representation and (ii) observational sampling and product construction. **In this case, the inconsistency arises primarily from the construction of the gridded OSNAP velocity product rather than from limitations in reanalysis resolution:** This anticyclonic eddy discussed by Lozier et al. (2017) was observed by a glider transect between June and November 2015 with distinct temperature and salinity characteristics. However, the OSNAP velocity field (as used here) is primarily determined by moorings spaced hundreds of kilometers apart and does not directly incorporate the glider-derived velocities. As a result, the OSNAP velocity field is not designed to resolve fine horizontal structures such as eddies located **between** moorings. E.g., in the region where the eddy was identified, the velocity reconstruction relies on the M3 and M4 dynamic height moorings to estimate geostrophic shear, which cannot represent the horizontal structure between them (see Fig. 9 in Lozier et al., 2017). Consequently, while the eddy is evident in the glider observations it is not retained in the monthly gridded OSNAP velocity product used here. The hydrographic property fields, by contrast, show broadly consistent behavior between the OSNAP product used here and the one from Lozier et al. (2017).

We edited L316 to make sure that this statement only applies to the velocity field, not the property field:

*However, no such signal is evident in the **velocity field of the publicly available OSNAP dataset used in this study, which is primarily determined using mooring observations that do not resolve the small-scale spatial structures such as eddies between moorings.***

We also clarified the wording in this section to make it clearer that we are talking about 2 separate events:

***As a separate example of mesoscale variability (unrelated to the 2015 heat transport peak), a strong anticyclonic eddy is also present in the western glider region**, clearly visible in GLORYS12V1 and, to a lesser extent, in GLORYS2V4.*

3. To add some clarity in the differences between the data used in this study, I suggest changing the structure of section 2 by adding a 'section 2.1 Data' that would introduce the data used in this study: OSNAP (including lines 101–106 currently in the following section), the reanalyses and the altimetry data. Sections 2.1 would become section 2.2 etc..

   We have restructured Section 2 accordingly by introducing a dedicated **Section 2.1 (Data description)**, which now explicitly introduces the observational OSNAP dataset, the ocean reanalyses, and the satellite altimetry data used in this study.

4. In the new section 2.1, I strongly recommend the authors discussing the differences and similarities between the reanalyses in terms of observations assimilated in these reanalyses. For example, can the authors clarify if the reanalyses assimilate OSNAP observations? Are they assimilating the same observations otherwise (e.g., Argo, altimetry, hydrographic sections…) meaning that their differences in OHT (or for example the results discussed at l.239-241) can be interpreted as a result of different horizontal resolutions and dynamical models only?

   We have expanded Section 2.1 to clarify both the similarities and differences between the reanalyses in terms of assimilated observations. We now explicitly state that none of the reanalyses assimilate direct velocity observations from the OSNAP array, or from current meters and ADCPs more generally. While temperature and salinity measurements from OSNAP may indirectly contribute via global in situ databases (which is difficult to verify), the OSNAP velocity field itself is not assimilated.

   We clarify that all reanalyses assimilate broadly similar observations, including in situ temperature and salinity profiles and sea level anomalies, which constrain the large-scale geostrophic circulation. However, they differ in horizontal

resolution, data assimilation schemes and in the spatial domain over which sea level anomalies are assimilated (e.g., ORAS5 versus the other products). To improve transparency, we have added a new table (Table 1) summarizing the assimilated data and assimilation schemes for each reanalysis. Therefore, differences in ocean heat transport among the reanalyses reflect a combination of model dynamics, resolution, and data assimilation methodology.

5. Finally, I recommend the authors discussing another OHT dataset produced by combining Argo, altimetry and gravimetry data from Calafat et al., 2025. https://doi.org/10.5194/os-21-2743-2025

We thank the reviewer for pointing us to this recently published and valuable dataset. The ocean heat transport estimates of Calafat et al. (2025) provide an important independent perspective on large-scale Atlantic heat transport variability.

However, this product provides heat transport estimates across complete latitude bands, whereas our study focuses on transport across the specific OSNAP section and its eastern and western components separately. Using the Calafat et al. dataset would therefore require approximating OSNAP by a zonal section, which we find introduces non-negligible differences compared to transports calculated along the actual OSNAP geometry (see figure below). In addition, the Calafat et al. product is provided at 3-monthly resolution and does not allow for a separation between OSNAP East and West, whereas the 2015 anomaly discussed here is most pronounced in OSNAP East.

[Figure]

*Fig.: Comparison of OHT calculated at the exact OSNAP coordinates (red) and across 60°N (green).*

For these reasons, we do not include this dataset in the quantitative analysis, but we now cite and briefly discuss it in the manuscript as an important complementary product for basin-scale heat transport assessments. We added the following in the manuscript:

*Calafat et al. (2025) present a novel Atlantic ocean heat transport dataset derived from a combination of Argo temperature profiles, satellite altimetry and gravimetric constraints. This product provides valuable insight into basin-scale heat transport variability across complete latitude bands. However, because it is defined along zonal sections and provided at 3-monthly resolution, it cannot be directly applied to the OSNAP section geometry or used to distinguish between OSNAP East and West, which is central to the present analysis. Nevertheless, such approaches offer an important complementary perspective on large-scale heat transport variability and are highly valuable for basin-scale assessments.*

**Minor Comments**

L. 62: use AMOC instead of MOC, as it was the authors' choice for the rest of the manuscript

Done

L. 108: consider using the term 'derived' instead of 'calculated'

Done

L. 114-115: Consider clarifying the grid that is used for the bilinear interpolation.

We have clarified the description of the interpolation:

*To assess cross-sectional biases and RMSE values between OSNAP and reanalyses, all reanalysis sections are interpolated bilinearly* **in the along-section and vertical directions onto the common OSNAP gridded section, defined by the OSNAP "along-section" distance coordinate and depth levels.**

L. 128-130: The sentences don't read properly. Maybe: 'Additionally, the potential temperature and a reference temperature are needed for estimating the heat transport. An unambiguous heat transports require closed volume transports, which is [...].'

To improve readability the sentence was changed to:

**Additionally, the potential temperature $\theta$ and a reference temperature $\theta\_ref$ are needed for estimating heat transports.**

L. 133: Can the authors describe in few sentences what is the StraitFlux's line integration method?

We have added a brief description of StraitFlux's line integration method:
*To avoid interpolation and preserve numerical conservation properties, net integrated transports from the reanalyses are calculated using StraitFlux's line-integration method,*

*in which transports are integrated directly along the native model grid cell faces that intersect the section, thereby approximating the target section as closely as possible on the native grid.*

L. 198: Typo 'Iceland basin'

Corrected

L. 203-205: EN4 includes a large number of Argo profiles and has probably a better spatial coverage over the subpolar North Atlantic than OSNAP that is missing observations in the basin interiors. However, there are possible issues of data QC in EN4. Consider also discussing in more details what structural and methodological uncertainties in OSNAP can explain these differences. From my understanding, OSNAP uses EN4 in the interior?

OSNAP does not use EN4 data directly in the basin interior. While OSNAP and EN4 may draw from overlapping in situ measurements (e.g. Argo profiles), they differ fundamentally in methodology: EN4 is an objective analysis with spatial smoothing, whereas OSNAP temperatures are derived from its dedicated observing system along the section and its own gridding methodology. Differences between the two likely reflect both EN4 mapping/QC choices and OSNAP uncertainties associated with sparse interior sampling and section construction.

L. 270: Related to my main comment #4, I recommend the authors to clarify in the new section 2.1 if they use independent data sources in the reanalysis.

At line 270, the term "independent data sources" refers to the multiple, distinct datasets used in the indirect (budget-based) heat transport estimates (listed in Table 1), including different atmospheric reanalyses, ocean heat content products, and sea-ice datasets, rather than data within the ocean reanalyses. We have slightly rephrased the sentence:

*This holds true across all combinations of datasets and both choke-point approaches, despite the use of* **multiple, independent data sources contributing to the indirect heat transport estimates (see Table 1).**

L. 290-292: I am confused, didn't the authors say that OSNAP cannot represent the circulation over this portion of the interior array because there aren't enough observations there?

We have clarified the text to explain that although the detailed circulation structure in the Iceland Basin is not resolved by OSNAP, the opposing transport branches largely compensate, resulting in consistent net accumulated transports between reanalyses and observations.

*This northward–southward circulation feature is not resolved by the OSNAP observations due to OSNAP's array design that relies on end-point dynamic height*

*moorings to capture the total integrated transport and its variability in the Iceland Basin.*
***However, because the opposing branches largely compensate each other, the net***
***accumulated heat transport over this segment is consistent between the***
***reanalyses and the OSNAP observations (Fig. 7d).***

L. 295: Consider clarifying if the potential uncertainties in these areas relates to uncertainties in the reanalyses or OSNAP observations?

We have clarified the text:

*More broadly, the reanalyses exhibit high variability (Fig. 7c) in regions that lack direct mooring* **observations in OSNAP, underscoring potential uncertainties in the OSNAP transport estimates in these areas.**

L.320: Is the barotropic compensation applied at OSNAP uniform in time or in space (horizontally and vertically) or both? How can it impact the correlation of OSNAP with SLA?

In OSNAP, the barotropic compensation is applied as a spatially uniform (horizontally and vertically throughout the whole section) but time-varying velocity offset to close the net transport budget at each time step. This spatially uniform compensating velocity is not expected to show tight pointwise correlations with SLA within this narrow glider section.

We have clarified this in the revised manuscript:

*We note a key methodological difference: OSNAP derives time-varying geostrophic shear from T/S and applies a* **spatially uniform, but time-varying barotropic compensation to obtain absolute velocities,** *whereas the reanalyses assimilate SLA and thus include time-varying surface geostrophic flow tied to SLA. Accordingly, the SLA -transport comparison below is intended as a diagnostic of reanalysis consistency and as context for OSNAP.* **In this framework, tight pointwise correlations between SLA gradients and OSNAP transport are not expected.** *Nevertheless, because along-section SLA gradients set the surface geostrophic shear, we'd expect a coherent relationship after spatial averaging over broader segments (e.g., the glider regions).*

L. 338-346: Related to my main comment #4, it would be easier to interpret this result with more details on the dataset assimilated by the different reanalyses. Do these two reanalyses (GLORYS2v4 and GLORYS12V1) assimilate the same SLA data? Why not showing the results for the other reanalyses?

GLORYS2V4 and GLORYS12V1 both assimilate multi-mission satellite altimetry derived sea level anomalies from CMEMS, but differ in horizontal resolution and assimilation methodology. We have clarified this in the text. We focus on these two products to highlight the impact of SLA assimilation and resolution on SLA-transport correlations. Other reanalyses are either not directly comparable in this context (e.g., ORAS5 does

not assimilate SLA north of 50°N) or show similar correlations and are omitted for clarity, as including all products would detract from the main focus of the study.

Revised paragraph:

*In contrast, GLORYS12V1 and GLORYS2V4 exhibit higher correlations with observed SLA gradients (0.40 and 0.39, respectively) and even larger values when compared to their own SLA fields (0.79 and 0.51).* **Both products assimilate the same multi-mission satellite altimetry-derived SLA observations but differ in horizontal resolution and assimilation methodology (see Table 1).** *Notably, the higher-resolution GLORYS12V1 shows the strongest correlations overall, consistent with its improved spatial representation of circulation features. Correlations are weaker for the ORAS5 reanalysis (not shown), with a correlation of just 0.25 against observed SLA gradients, likely a result of it not assimilating sea level anomalies north of 50°N.*

L. 347-355: Can the authors clarify why smooth all fields over 1deg resolution while the coarser resolution from reanalysis or altimetry is 1/4deg? In my view, only GLORYS12V1 should be smoothed at 1/4deg.

We thank the reviewer for raising this point. We realize that the original wording was misleading. The 1° smoothing is **not applied to the data prior to the calculation of correlations**. All SLA-transport correlations are computed on the native grids/sections of the respective datasets.

The 1° smoothing is applied **only to the along-section correlation curves shown in Fig. 11**, purely for visualization purposes, in order to reduce small-scale noise and improve readability of the plotted results. The reported correlation values are based on the unsmoothed fields.

We have revised the manuscript to clarify this distinction and avoid confusion regarding the role of smoothing:

**To reduce small-scale noise and facilitate visualization, the along-section correlation curves shown in Fig. 11 are smoothed to 1° resolution. This smoothing is applied only for plotting purposes and does not affect the calculation of the correlations themselves, which are performed on the native-resolution fields along the OSNAP section.** *This allows us to focus on mesoscale dynamics while suppressing small-scale variability and potential sampling mismatches. Despite this visual smoothing, pointwise correlations across the section remain relatively low for the OSNAP dataset.*